# SNX19 Interacts with Caveolin-1 and Flotillin-1 to Regulate D_1_R Endocytosis and Signaling

**DOI:** 10.3390/biomedicines13020481

**Published:** 2025-02-15

**Authors:** Bibhas Amatya, Jacob Q. M. Polzin, Van A. M. Villar, Jiang Yang, Prasad Konkalmatt, Xiaoyan Wang, Raisha C. Cadme, Peng Xu, John J. Gildea, Santiago Cuevas, Ines Armando, Robin A. Felder, Pedro A. Jose, Hewang Lee

**Affiliations:** 1Division of Kidney Diseases & Hypertension, Department of Medicine, The George Washington University School of Medicine & Health Sciences, Washington, DC 20052, USA; bibhasamatya78@gwmail.gwu.edu (B.A.); polzin@gwmail.gwu.edu (J.Q.M.P.); vvillar@email.gwu.edu (V.A.M.V.); pkonkalmatt@gmail.com (P.K.); xiaoyan51@hotmail.com (X.W.); raisha.campisi@email.gwu.edu (R.C.C.); santiago.cuevas@imib.es (S.C.); iarmando@email.gwu.edu (I.A.); pjose01@email.gwu.edu (P.A.J.); 2Department of Medicine, University of Maryland School of Medicine, Baltimore, MD 21201, USA; 3Division of Nephrology, Department of Medicine, University of Maryland School of Medicine, 20 Penn Street, HSF II, Baltimore, MD 21201, USA; yangjianac@163.com; 4Department of Clinical Nutrition, The Third Affiliated Hospital of Chongqing Medical University, Chongqing 400016, China; 5Department of Nephrology, BenQ Medical Center, The Affiliated BenQ Hospital of Nanjing Medical University, Nanjing 210029, China; 6Department of Pathology, University of Virginia Health Sciences Center, Charlottesville, VA 22908, USA; px3x@virginia.edu (P.X.); jjg5b@virginia.edu (J.J.G.); raf7k@virginia.edu (R.A.F.); 7Physiopathology of the Inflammation and Oxidative Stress Laboratory, Molecular Inflammation Group, Biomedical Research Institute of Murcia Pascual Parrilla (IMIB), 30120 Palmar, Spain; 8Department of Pharmacology & Physiology, The George Washington University School of Medicine & Health Sciences, Washington, DC 20052, USA

**Keywords:** caveolin-1, dopamine D_1_ receptor, flotillin-1, endocytosis, lipid raft, SNX19

## Abstract

**Background:** Sorting nexin 19 (SNX19) is important in the localization and trafficking of the dopamine D_1_ receptor (D_1_R) to lipid raft microdomains. However, the interaction between SNX19 and the lipid raft components caveolin-1 or flotillin-1 and, in particular, their roles in the cellular endocytosis and cell membrane trafficking of the D_1_R have not been determined. **Methods:** Caveolin-1 and flotillin-1 motifs were analyzed by in silico analysis; colocalization was observed by confocal immunofluorescence microscopy; protein-protein interaction was determined by co-immunoprecipitation. **Results:** In silico analysis revealed the presence of putative caveolin-1 and flotillin-1 binding motifs within SNX19. In mouse and human renal proximal tubule cells (RPTCs), SNX19 was localized mainly in lipid rafts. In mouse RPTCs transfected with wild-type (WT) *Snx19*, fenoldopam (FEN), a D_1_-like receptor agonist, increased the colocalization of SNX19 with caveolin-1 and flotillin-1. FEN also increased the co-immunoprecipitation of SNX19 with caveolin-1 and flotillin-1, effects that were prevented by SCH39166, a D_1_-like receptor antagonist. The FEN-mediated increase in the residence of SNX19 in lipid rafts and the colocalization of the D_1_R with caveolin-1 and flotilin-1 were attenuated by the deletion of a caveolin-1 (YHTVNRRYREF) (ΔCav1) or a flotillin-1 (EEGPGTETETGLPVS) (ΔFlot1) binding motif. The FEN-mediated increase in intracellular cAMP production was also impaired by the deletion of either the flotillin-1 or caveolin-1 binding motif. Nocodazole, a microtubule depolymerization inhibitor, interfered with the FEN-mediated increase in the colocalization between SNX19 and D_1_R. **Conclusion:** SNX19 contains caveolin-1 and flotillin-1 binding motifs, which play an important role in D_1_R endocytosis and signaling.

## 1. Introduction

The dopamine D_1_ receptor (D_1_R), one of the two dopamine D_1_-like receptors (D_1_R and D_5_R) in mammals, is a G protein-coupled receptor (GPCR) that is highly expressed in the kidney [1,2]. Upon binding to dopamine, the D_1_R at the plasma membrane undergoes a conformational change and activates adenyl cyclase, increasing the production of the second messenger cyclic AMP (cAMP) to drive downstream signaling and trafficking processes [1,2]. Similar to other GPCRs, the endocytosis of the D_1_R is fundamental for its precise intracellular spatiotemporal localization, downstream signaling, and eventual function [3].

Lipid rafts (LRs) are specialized membrane microdomains that are enriched in sphingolipids and cholesterol [4]. LRs serve as organizing centers for the assembly of signaling molecules and the creation of compartmentalized platforms capable of assembling the signal transduction machinery [5,6]. Moreover, LRs facilitate protein–protein and protein–lipid interactions at the plasma membrane and regulate GPCR signal transduction, internalization, and intracellular trafficking [5,6], e.g., the D_1_R [6,7].

Caveolae are important components of the machinery that regulates the compartmentalization of LRs and GPCR signaling [4,6,7,8]. Caveolin-1, a major constituent of caveolae, has various functions related to endosomal membrane trafficking and receptor signal transduction [4,6,7,8]. For example, caveolin-1 is necessary for D_1_-like receptor-mediated internalization of Na^+^/K^+^-ATPase in renal proximal tubule cells (RPTCs) [9,10]. Inhibition of renal caveolin-1 or chronic disruption of LRs impairs renal sodium excretion [9]. Flotillin-1, another LR component [4,8], is also involved in plasma membrane endocytosis [11,12] and GPCR-mediated intracellular signaling [13].

We have reported that sorting nexin 19 (SNX19), an adaptor protein, is important in the targeting of the D_1_R into lipid raft microdomains [14]. SNX19 is ubiquitously found in the plasma membrane and cytoplasm, specifically in endosomes, the endoplasmic reticulum, lipid droplets, lysosomes, and mitochondria [14,15,16,17]. The diverse intracellular distribution of SNX19 implies multiple cellular functions [14,15,16,17]. In mice, siRNA-mediated silencing of *Snx19* mistargets the D_1_R to non-LRs [14]. Therefore, SNX19 is essential for the lipid raft residence of D_1_Rs [14,15]. However, the role of caveolin-1 and flotillin-1 in the SNX19-mediated D_1_R signaling and trafficking in RPTCs is not known.

In this study, we identified putative caveolin-1 and flotillin-1 binding motifs within the SNX19 primary structure and found that both are necessary for SNX19-mediated D_1_R endocytosis and signaling in RPTCs.

## 2. Materials and Methods

### 2.1. Plasmids, Antibodies, and Reagents

Plasmids of human wild-type (WT)-SNX19 (Accession: NM_014758), ∆Cav1-SNX19 (deleted bp: +1645 ttccacccatacacactctatactgtgaagtacgag), and ∆Flot1-*SNX19* (deleted bp: +1273 gaggagggtccagggacagaaacagagacaggcctgccggtctcc) were purchased from GenScript (Piscataway, NJ, USA). Our in-house affinity-purified D_1_R polyclonal antibody (clone 408) has been described and verified [14,18,19,20,21]. Anti-DYK tag antibodies for detecting transfected SNX19 were purchased from GenScript. Anti-SNX19 antibodies were purchased from Antibodies-online (Cat. No. ABIN1343958; Limerick, PA, USA) and Millipore Sigma (Cat. No. HPA013447; Burlington, MA, USA). Antibodies against caveolin-1 (Cat. No. ab192869 and ab36152) and flotillin-1 (Cat. No. ab133497 and ab13493) were purchased from Abcam (Waltham, MA, USA). Methyl-β-cyclodextrin (M-β-CD) and other reagents, unless otherwise stated, were purchased from Sigma-Aldrich.

### 2.2. Cell Culture and Transfection

Mouse RPTCs were purchased from American Type Culture Collection (ATCC) (Manassas, VA, USA) and cultured in a 1:1 mixture of Dulbecco’s modified Eagle’s medium (DMEM) and Ham’s F-12 medium (Invitrogen, Frederick, MD, USA), supplemented with 5% fetal bovine serum, selenium (5 ng/mL), insulin (5 µg/mL), transferrin (5 µg/mL), dexamethasone (5 ng/mL), triiodothyronine (4 pg/mL), and epidermal growth factor (10 ng/mL).

The source of the human RPTCs and their culture have been reported [9,10,14,18].

Constructs of WT-SNX19, ∆Cav1-SNX19, and ∆Flot1-SNX19 were transfected into mouse RPTCs grown in 6-well plates, using FuGene HD transfection reagent (Promega, Madison, WI, USA), according to the manufacturer’s instructions and our published procedure [14]. The expression of the transfected SNX19 constructs was determined by Western blotting (Appendix A) and quantitative real-time polymerase chain reaction (qRT-PCR) (Appendix A). Plasmid D_1_R-GFP has been previously reported [14]. Specific *SNX19* and mock siRNAs were purchased from Qiagen (Germantown, MD, USA). Human RPTCs, grown in 6-well plates, were co-transfected with plasmid D_1_R-GFP (0.5 µg/well) and mock or specific *SNX19* siRNA (20 nmol/L) for 48 h. Efficiency of SNX19 mRNA and protein expressions by specific *SNX19* siRNA after transfection were determined by qRT-PCR and immunoblotting, respectively (Appendix A).

### 2.3. Isolation of LR Fractions

We have reported the isolation of LR fractions using sucrose gradient ultracentrifugation [14,20,22]. Human RPTCs, grown to confluence, were treated with vehicle or methyl-β-cyclodextrin (M-β-CD, 2%) for 1 h at room temperature and then harvested and washed twice with ice-cold phosphate-buffered saline (PBS). The cell pellets were lysed by a Dounce homogenizer in ice-cold SPE Buffer I (FOCUS SubCell kit, G-Biosciences, St Louis, MO, USA). The cell lysates were centrifuged at 700× *g* for 10 min at 4 °C. The supernatants were then centrifuged at 12,000× *g* for 15 min at 4 °C. The remaining supernatants were subsequently centrifuged at 100,000× *g* for 60 min at 4 °C in a SW50.1 swinging bucket rotor (Beckman Coulter, Palo Alto, CA, USA). The membrane-enriched pellets were subjected to detergent-free sucrose gradient ultracentrifugation. After ultracentrifugation, twelve 1 mL fractions were collected. The fractionated proteins were boiled with Laemmli buffer and subjected to immunoblotting [14,20,22].

### 2.4. Immunoblotting

The protein samples, containing a protease and phosphatase inhibitor cocktail (Thermo Fisher Scientific, Rockford, IL, USA), were separated by sodium dodecyl sulfate-polyacrylamide gel electrophoresis (SDS-PAGE) and transferred onto nitrocellulose membranes, using the Trans-Blot Turbo apparatus (Bio-Rad, Hercules, CA, USA) [18,19]. The membranes were blocked and then probed with primary and secondary antibodies in Odyssey Blocking Buffer (Li-Cor Biosciences, Lincoln, NE, USA). The images were visualized by a LiCor Odyssey Imaging system (Version: 3.0.21).

### 2.5. Cyclic AMP Assay

Cyclic AMP (cAMP) was assayed using an immunoassay kit (Arbor Assays, Ann Arbor, MI, USA) [18,19]. Briefly, mouse RPTCs were grown in 12-well plates. The RPTCs at ~75% confluence were pretreated with the phosphodiesterase inhibitor 3-isobutyl-1-methylxanthine (IBMX; 1 M; Sigma-Aldrich, St. Louis, MO, USA) before the addition of vehicle, fenoldopam (FEN) (25 nM, 30 min), or nocodazole (10 µM, 1 h). The supernatants were collected for cAMP assay following the manufacturer’s instructions and the optical density was measured at 450 nm by an ELISA plate reader.

### 2.6. Confocal Fluorescence Microscopy and Colocalization Analysis

Confocal fluorescence microscopy was performed, as previously described [14,22]. Briefly, the RPTCs were fixed with 4% paraformaldehyde in PBS for 20 min at room temperature. After washing with PBS, the fixed cells on coverslips were incubated with primary antibodies overnight at 4 °C. The coverslips were then incubated with the proper Alexa Fluor-488 and -555 secondary antibodies for 2 h at 4 °C. The coverslips were mounted in an antifade mounting medium (Vectashield, Burlingame, CA, USA) and sealed onto glass slides. The samples were imaged with a Zeiss, Oberkochen, Germany, LSM 710 confocal laser scanning microscope, equipped with a Plan-Apochromat 63x/1.40 NA oil-immersion objective. To ensure that the image was within the intensity range of the detector, the laser intensity, brightness (offset), and contrast (gain) were carefully set by monitoring the display of fluorescence histograms.

The colocalization of two labels was analyzed using ImageJ (Version 1.54d, NIH, Bethesda, MD, USA) with JACoP colocalization algorithms [23,24]. The confocal images for the two labels were opened and assigned as Image A (green channel) and Image B (red channel). Thresholding was carefully performed to eliminate the background pixels for both fluorophores. Because the quantification of colocalization varies due to multiple factors, including intensity, background, and microscope configuration, two independent measures, Pearson’s correlation coefficient and Manders’ overlap coefficient, were applied to each measurement. Pearson’s method quantifies the linear correlation between two channel images with a range from −1 (complete exclusion) to +1 (complete colocalization). Manders’ method quantifies colocalization based on the intensities of the pixels in each channel image with a range from 0 (no overlap) to 1 (complete overlap). Although the values of the two methods correlated well with each other, we reported the overlap values of one channel over another channel using the Manders’ method, taking into consideration that the single Pearson’s value does not distinguish individual channels and is affected by the background [24,25].

The colocalization images were generated as previously described [22]. Threshold values were manually tuned to eliminate the background pixels. The colocalization images were generated following the procedure of the program IMARIS 9.8 version (Bitplane AG, Saint Paul, MN, USA). [22].

### 2.7. Statistical Analysis

Results are expressed as mean ± standard deviation (SD). Significant differences among groups (n > 2) were determined by ANOVA, followed by Newman–Keuls or Holm–Sidak post hoc test, and between two groups by Student’s *t* test. *p* < 0.05 was considered statistically significant (GraphPad Prism 10.0, La Jolla, CA, USA).

## 3. Results

### 3.1. SNX19 Is Present in Lipid Rafts (LRs)

We have reported that in human and mouse RPTCs, SNX19 facilitates the palmitoylation of D_1_Rs, which promotes the partitioning of D_1_Rs into LRs [14], where other signaling components are located. Therefore, we studied in more detail the expression of SNX19 in LRs in RPTCs. Sucrose gradient fractionation of human RPTC membranes showed that SNX19 was distributed both in buoyant fractions, known as LRs (fractions 1–6), and less buoyant fractions, known as non-LRs (fractions 7–12) (Figure 1A). The cholesterol-depleting drug methyl-β-cyclodextrin (M-β-CD), which disrupts LRs [20], redistributed the majority of SNX19 from the buoyant fractions to the less buoyant fractions (Figure 1A). This shift in the distribution of SNX19 caused by M-β-CD confirmed that SNX19 resides in both LRs and non-LRs.

Similar to the distribution of SNX19 in both LR and non-LR fractions of cell membranes, the D_1_R was also distributed in both LR and non-LR fractions in RPTCs (Figure 1A). M-β-CD increased its distribution to non-LR fractions, which confirmed that D_1_R resides in both LRs and non-LRs and is suggestive of a possible interaction between SNX19 and the D_1_R in LRs and non-LRs.

Confocal images showed that in mouse RPTCs, SNX19 colocalized with cholera toxin B-subunit (CTxB), a commonly used LR marker [14] (Figure 1B). The colocalization of SNX19 and CTxB was disrupted by M-β-CD (Figure 1B), confirming the residence of SNX19 in LRs in mouse RPTCs, which is consistent with the sucrose gradient fractionation results in human RPTCs (Figure 1A).

### 3.2. SNX19 Mediates D_1_R Endocytosis in Human and Mouse RPTCs

Endocytosis is an important mechanism in the regulation of receptor trafficking and signaling [3,15]. Considering the presence of both SNX19 and D_1_R in LRs, we investigated the potential interaction between SNX19 and the D_1_R on D_1_R endocytosis in RPTCs which endogenously express both SNX19 and D_1_R. In human RPTCs co-transfected with D_1_R-GFP plasmid and non-silencing or specific *SNX19* siRNA, live cell imaging showed that the D_1_-like receptor agonist FEN [14,22] caused D_1_R endocytosis (Figure 2). By contrast, D_1_R remained at the surface of the human RPTCs that were transfected with specific *SNX19* siRNA. These data indicate that SNX19 plays a critical role in the endocytosis of D_1_R in human RPTCs.

Rab5, a key component of the early endocytosis machinery, is widely used as an early endocytosis marker [26]. Consistent with the role of SNX19 in human D_1_R trafficking, SNX19 increased the localization of the D_1_R in Rab5-positive early endosomes in mouse RPTCs transfected with wild-type (WT) SNX19 plasmids, compared with those transfected with plasmids not expressing SNX19 (Figure 3). We have reported that the endocytosis of the D_1_R is increased by the D_1_-like receptor agonist FEN in human RPTCs [14]. These data indicate that SNX19 mediates D_1_R endocytosis in both human and mouse RPTCs.

### 3.3. LR Residence of SNX19 Is Dependent on Its Caveolin-1 and Flotillin-1 Binding Motifs

Both caveolin-1 and flotillin-1 are LR components that mediate plasma membrane receptor endocytosis [8]. Therefore, we studied the colocalization of SNX19 with caveolin-1 and flotillin-1 in RPTCs. The in silico analysis showed that human and mouse SNX19 contains both caveolin-1 like binding motif [27,28] (YHTVNRRYREF) (Figure 4A) and flotillin-1-like binding motif [29,30] (EEGPGTETETGLPVS) (Figure 4B).

We transfected human wild-type (WT) SNX19 into mouse RPTCs and found that some WT-SNX19 resides in the LRs marked by CTX-B (Figure 4C). The localization of SNX19 in LRs increased after treatment with the D_1_-like receptor agonist [9,10,14,18,19,20,22] FEN (25 nM, 30 min), as shown by the increase in the colocalization of SNX19 with CTxB (Appendix A). The FEN-mediated increase in the colocalization of the D_1_R and SNX19 was blocked by SCH39166 (SCH, 1 µM) (Appendix A), a D_1_-like receptor antagonist [22], which by itself had no effect (Appendix A). However, the deletion of either the caveolin-1 or flotillin-1 binding motif within SNX19 impaired the FEN-mediated increase in the residence of SNX19 in LRs in mouse RPTCs (Figure 4C,D), indicating that the residence of SNX19 in LRs is dependent on both its caveolin-1 and flotillin-1 binding motifs.

To evaluate the potential interaction of SNX19 with caveolin-1 and flotilin-1, co-immunoprecipitation studies were performed. In the basal state, SNX19 co-immunoprecipitated with caveolin-1 (Figure 5A) and flotillin-1 (Figure 5B). This interaction was also present in reverse conditions, i.e., immunoprecipitation with anti-caveolin-1 or anti-flotillin-1 and immunoblotting with anti-SNX19 (Figure 5). FEN (25 nM, 30 min) increased the co-immunoprecipitation of SNX19 with caveolin-1 and flotillin-1 (Figure 5). In addition, the treatment of the human RPTCs with a D_1_-like receptor antagonist [22], SCH (1 µM, 30 min), which by itself had no effect, prevented FEN from increasing the co-immunoprecipitation of SNX19 with caveolin-1 or flotillin-1 (Figure 5). These results show that the association of SNX19 with both caveolin-1 and flotillin-1 is regulated by D_1_-like receptors in human RPTCs.

### 3.4. SNX19-Mediated D_1_R Endocytosis Is Dependent on Both Caveolin-1 and Flotillin-1 Binding Motifs

We next studied the importance of caveolin-1 and flotillin-1 binding motifs in SNX19-mediated endocytosis. FEN (25 nM, 30 min) increased the colocalization of D_1_R with caveolin-1 in WT-SNX19 (Figure 6A) and ΔFlot1-SNX19 (Figure 6C) but not in ΔCav1-SNX19-transfected (Figure 6B) mouse RPTCs. Conversely, FEN (25 nM, 30 min) increased the colocalization of D_1_Rs with flotillin-1 in WT- and ΔCav1-SNX19-transfected mouse RPTCs but to a lesser extent in ΔFlot1-SNX19-transfected mouse RPTCs (Appendix A). These findings indicate that both caveolin-1 and flotillin-1 binding motifs are important in SNX19-mediated D_1_R caveola-dependent endocytosis.

### 3.5. SNX19-Mediated D_1_R Signaling Is Dependent on Both Caveolin-1 and Flotillin-1 Binding Motifs

We next studied D_1_-like-receptor-mediated cAMP production in the WT-, ΔCav1-, and ΔFlot1-SNX19-transfected mouse RPTCs. FEN (25 nM, 30 min) increased cAMP production in WT-SNX19-transfected mouse RPTCs. However, the FEN-mediated increase in cAMP production was markedly decreased in either ΔCav1- or ΔFlot1-SNX19-transfected mouse RPTCs (Appendix A). These data indicate that the SNX19-mediated D_1_-like receptor cAMP signaling in LRs requires both caveolin-1 and flotillin-1 binding motifs.

### 3.6. SNX19-Mediated D_1_R Regulation Requires Microtubule Integrity

Microtubules play an important role in LR signaling and trafficking in polarized epithelial cells [31,32]. Therefore, we next examined their potential roles in the SNX19-mediated D_1_R regulation. Nocodazole, a microtubule polymerization inhibitor [33], aggregated β-tubulin filaments and disrupted the normal SNX19 plasma membrane distribution (Appendix A). Nocodazole interfered with the FEN-mediated increase in the colocalization of SNX19 with D_1_R (Figure 7). However, it did not affect the D_1_-like-receptor-mediated increase in cAMP production (Figure 8), demonstrating that microtubule polymerization does not significantly affect the cAMP signaling process. This may occur because microtubules are not involved in the plasma membrane signaling machinery of the D_1_R (G proteins, adenylyl cyclase, etc.) but play a role in the subsequent SNX19-mediated D_1_R trafficking, which needs further investigation.

## 4. Discussion

Upon binding to its ligands, the D_1_R, like other GPCRs, is activated, internalized, and sorted into the early endosomal network, via endocytosis [2,3]. The D_1_R undergoes endocytosis through clathrin-mediated [18,34] and non-clathrin-mediated pathways [35]. Non-clathrin-mediated pathways are mainly LR-dependent [8]. In clathrin-mediated endocytosis, the D_1_R is inserted into clathrin-coated endocytic vesicles [18]. In LR-mediated endocytosis, GPCRs begin their internalization with the formation of a flask-shaped caveola, an invaginated plasma membrane, which contains caveolin-1 and flotillin-1 [6,8,32,36]. In the current study, we found that SNX19 colocalized and co-immunoprecipitated with both caveolin-1 and flotillin-1 in RPTCs, showing the association of SNX19 with both caveolin-1 and flotillin-1 in LRs. Their association was increased by FEN, a D_1_-like receptor agonist [9,10,14,18,19,20,22].

SNX19 is a member of the SNX-PXA-RGS-PXC subfamily of sorting nexins that play diverse roles in GPCR signaling and trafficking [14,15]. This subfamily, which includes SNX13, SNX14, SNX19, and SNX25, has a central phox homology (PX) domain that is flanked by N-terminal PXA and C-terminal PXC domains [15,17]. The PX domain of SNX19, like other SNX proteins, mediates its role in cell signaling and membrane trafficking [14,15,37,38]. SNX19 is widely expressed in many organs, including the bone marrow, brain, heart, kidney, lung, and pancreas, and ubiquitously distributed in the cytoplasm and plasma membrane, including the sub-plasma membrane area [15,16,17,37,38,39], which is consistent with its diverse cellular functions. Indeed, our current study demonstrated a large amount of SNX19 residing in LRs of the plasma membrane that is associated with its regulation of D_1_R signaling and endocytic trafficking in RPTCs.

SNX19 plays an important role in membrane receptor signaling and cell membrane trafficking by regulating protein–protein interactions. The first identified SNX19 interactor is islet antigen 2, a major autoantigen in type 1 diabetes, which is involved in the regulation of insulin secretion by pancreatic β cells [40] and dopamine release in PC12 cells [41]. SNX19 interacts with the D_1_R, regulating its palmitoylation and promoting its lipid raft partition in RPTCs [14]. In this study, we identified two motifs that putatively bind to caveolin-1 and flotillin-1 within the SNX19 protein sequence. Caveolin-1, a major component of the caveola that is important in caveolar biogenesis [42], has a caveolin binding motif within its interactors. The canonical caveolin binding motif has a consensus sequence φxφxxxxφxxφ [27,28], with some variations (where φ is an aromatic amino acid and x is any amino acid) among different interactors [43,44]. Human SNX19 has six residues between two aromatic residues, Tyr 576 and Tyr 583, that are slightly different from human potassium voltage-gated channel Kv1.3 with four residues between two aromatic amino acids Phe (at 216) and Trp (at 221) [28]. Gαq directly interacts with caveolin-1 determined by fluorescence resonance energy transfer (FRET) [45]. The interaction of caveolin-1 with G protein subunits, e.g., Gαs, Gαi2, and Gαo was shown by GST pull-down experiments [27] and proteomics with subdiffraction-limit microscopy [46]. Gαi2 has canonical a caveolin binding motif with four amino acids between two aromatic amino acids Phe (at 192 and 197) (Figure 4A), whereas Gαs and Gαq have a non-canonical caveolin binding motif with six residues between two aromatic amino acids Phe (at 212 and 219 and at 194 and 201, respectively) (Figure 4A). Whether or not the canonical caveolin-1 binding motif within SNX19 is strictly required for its binding to caveolin-1 or caveolin-1 binds to another region of SNX19 warrants further investigation. Both Kv1.3 [28] and Gαq [45] are established interactors of caveolin-1. Flotillin-1, another important component of LRs, assembles to form molecular scaffolds to regulate cellular signaling and membrane trafficking [47]. The sorbin homology (SoHo) domain, a sequence with similarity to the peptide sorbin, has been identified to bind to flotillin-1, which serves as an adapter protein to link with LRs [29,30]. The sequence of the SoHo domain of SNX19 (residues 425–439) has high similarity (Figure 4B) to human and mouse ArgBP2A, vinexin, and c-Cbl-associated protein (CAP), which are established flotillin-1 binding proteins [29,30].

Deletion of either the caveolin-1 or flotillin-1 binding motif impaired the FEN-mediated increase in the residence of SNX19 in LRs and the SNX19-mediated D_1_R endocytosis. These suggest that these two motifs are required for the residence of SNX19 in LRs and the regulation of D_1_R by SNX19 in RPTCs. Depending on the type of cell and tissue, flotillin-1 can be in the same caveola as caveolin-1 [11,48,49] or in a non-caveolar LR microdomain distinct from caveolin-1 [50,51]. Further studies are needed to determine whether SNX19 binds to the same caveolar microdomain through caveolin-1 and flotillin-1 binding motifs in RPTCs. Caveolin-1 palmitoyl tail was found to increase its binding with cholesterol-rich membrane bilayers in a coarse-grain simulation experiment [52]. Palmitoylation of flotillin-1 also favors its binding to LRs and regulation of membrane trafficking and cellular signaling [53]. In human RPTCs, SNX19 is important for D_1_R palmitoylation and lipid raft localization [14]. However, additional studies are needed to determine whether SNX19 plays a role in the palmitoylation of caveolin-1 and flotillin-1 and what enzymes are involved in such effects.

Microtubules and their associated motors promote plasma membrane tubulation and GPCR endosomal trafficking [54]. CTxB, a caveola marker, is located within microtubule-dependent tubular invaginations [55], indicating that microtubules may play important roles in caveola-mediated GPCR endocytosis. The α_2B_-adrenergic receptor interacts with tubulins, and disruption of its interaction by mutation of the Arg residues interferes with its transport from the endoplasmic reticulum to the plasma membrane [56]. Metabotropic glutamate receptors also interact with tubulin [57], indicating the microtubule’s role in the regulation of presynaptic trafficking of glutamate receptors. KIF16B, a microtubule motor kinesin protein, interacts with endosome lipid phosphatidylinositol 3-phosphate via its PX domain and regulates receptor recycling or degradation [58], consistent with the notion that microtubules play important roles in membrane receptor endocytic trafficking. SNX19, through its N-terminal transmembrane, tethers to the endoplasmic reticulum [16]. The fact that microtubules closely align with endoplasmic reticula, and the endoplasmic reticulum morphology, distribution, and dynamics in SNX19 knockout cells are similar to that of SNX19-positive cells [16] indicate that microtubules play a role in the transient contacts between endoplasmic reticulum and endosomal system. In our study, SNX19 colocalized with β-tubulin and microtubule polymerization inhibition impaired the FEN-mediated SNX19 colocalization with the D_1_R. Actin, another cytoskeleton, along with microtubules, also plays a crucial role in GPCR spatial dynamic distribution in the plasma membrane and endocytic compartments [54,55,56,57,58]. The role of cytoskeletons and their motor proteins in SNX19-mediated D_1_R trafficking needs further investigation.

Endocytosis of a GPCR is intimately intertwined with its signaling [2,3]. Agonist-GPCR interaction triggers G protein-mediated signaling and endocytic trafficking modulates the amount of receptors in the plasma membrane and thereby regulates the extent of signaling in the long-term. In human RPTCs, FEN (short-term) increases D_1_R and caveolin-1 redistribution the plasm membranes [10]. In HEK-293 cells heterologously expressing human D_1_R (D_1_R-HEK293 cells), FEN increases the translocation of caveolin-2 and flotillin-1 in lipid rafts and the association of caveolin-2 with D_1_R [59], which is blocked by SCH23390, a D1 like receptor antagonist [59]. In C57BL/6 mice, renal silencing *Snx19* in mice decreases the expression of D_1_R with decrease in SNX19 expression [14]. Endosomes serve as intracellular signaling sites after receptor endocytosis [3,60]. The PXA-RGS-PX-PXC subfamily, except SNX19, contains a regulator G protein signaling (RGS) domain that plays a role in the attenuation of GPCR function and negatively regulates G protein-related signaling cascades. The RGS domain in SNX proteins, like canonical RGS proteins, attenuates GPCR and related G protein signaling [61]. However, SNX19, which lacks the RGS domain, can still promote D_1_R signaling in LRs domain at the plasma membrane [14,15]. Moreover, the deletion of caveolin-1 or flotillin-1 binding motif within SNX19 attenuates D_1_R residence in LRs and cAMP production in mouse RPTCs (current study). In addition to D_1_R, SNX19 also regulates the signaling of histamine receptor H4, a GPCR that is important in the initiation and maintenance of inflammation in mouse lungs, following ammonia exposure [62]. SNX19 also binds to PtdIns(4,5)P2 or PtdIns(3,4,5)P3 and decreases the phosphorylation of Akt/PKB leading to downregulation of Akt signaling under high glucose conditions [39]. Such findings indicate that the RGS domain is not always necessary for SNX-mediated regulation of GPCR signaling.

SNX19-mediated D_1_R endocytosis and signaling is associated with renal blood pressure regulation. The renal subcapsular infusion of specific *Snx19* siRNA increases blood pressure in C57BL/7 mice [14]. Renal-restricted disruption of lipid rafts by M-β-CD decreases urinary sodium excretion [14] and caveolin-1 plasma membrane expression [63]. Caveolin-1 and lipid rafts are necessary for normal D_1_-like-receptor-dependent internalization of Na^+^/K^+^-ATPase in human RPTCs [10]. Activation of the D_1_R decreases membrane expression and activities of type IIa sodium phosphate co-transporter (NaP IIa), sodium hydrogen exchanger-3 (NHE3), and Na^+^/K^+^-ATPase in the renal proximal tubule [10,63,64,65]. Whether or not SNX19 regulates NaP IIa, NHE3, and Na^+^/K^+^-ATPase in the microdomain of the plasma membrane needs further investigation.

## 5. Conclusions and Perspective

This study demonstrated that SNX19 resides in LRs in RPTCs. The D_1_-like receptor agonist FEN increases the colocalization of SNX19 with caveolin-1 and flotillin-1 in LRs, which is attenuated by the deletion of a caveolin-1 or a flotillin-1 binding motif. In addition, deletion of either the caveolin-1 or flotillin-1 binding motif impairs the FEN-mediated cAMP production. Nocodazole, a microtubule depolymerization inhibitor, impairs SNX19-mediated D_1_R regulation. SNX19 contains caveolin-1 and flotillin-1 binding motifs, which play an important role in D_1_R endocytosis and signaling.

Both caveolin-1 and flotillin-1 selectively serve as a scaffold for GPCR, including D_1_R, Gαs protein, and their known and unknown effectors in membranes, and have been implicated in the regulation of a variety of cellular functions, including endocytosis, exocytosis, and the transmembrane signaling cascade. The discovery of caveolin-1 and flotillin-1 binding motifs within SNX19 in this study lays the foundation for further deciphering the mechanisms of the SNX19 regulation of sodium transport in the renal proximal tubule and therefore blood pressure. The diversity of SNX19 binding with caveolin-1 and flotillin-1 could be studied by cryogenic electron microscopy [66,67] and can reveal the dynamic D_1_R signaling and trafficking in the regulation of sodium transport regulated by SNX19. Targeting the binding sites of SNX19 with caveolin-1 and flotillin-1 could be a new potential therapeutic approach to hypertension.

## Figures and Tables

**Figure 1 biomedicines-13-00481-f001:**
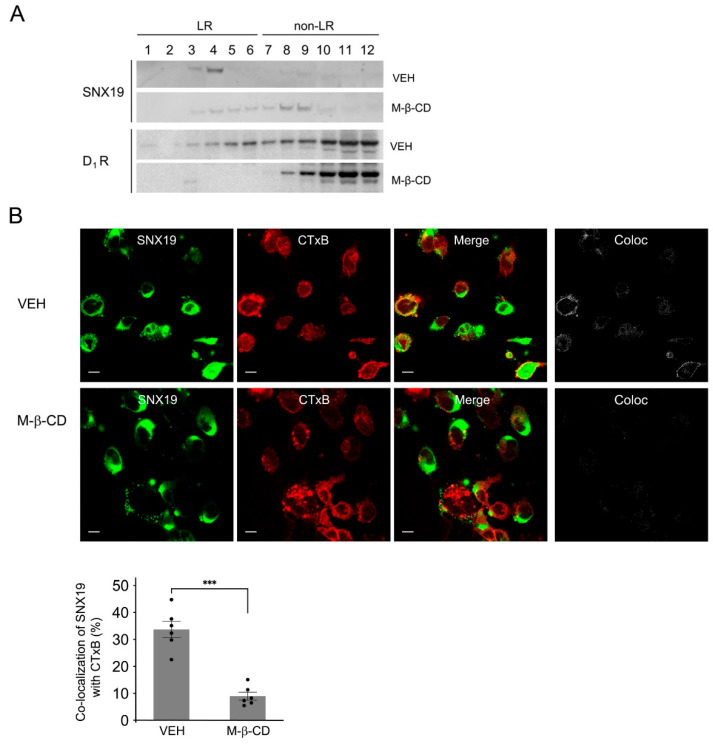
Residence of SNX19 in LRs in human and mouse renal proximal tubule cells (RPTCs). (**A**) Human RPTCs were treated with vehicle (VEH) or methyl-β-cyclodextrin (M-β-CD), a lipid raft (LR) disruptor. The cell lysates were subjected to sucrose gradient fractionation. The fractions were immunoblotted with antibodies against SNX19 (upper panel) and D_1_R (lower panel), as indicated. (**B**) Mouse RPTCs were fixed with 4% paraformaldehyde and stained with anti-DYK (for SNX19, green) and cholera toxin B-subunit (CTxB, red); yellow in the merged images shows the colocalization of SNX19 with CTxB. Images are representative of three independent experiments. Bar, 10 µm. A panel of separate colocalization images in “Coloc” were generated as described in Methods section. Coloc = colocalization. Quantification of the images of vehicle (VEH)- and M-β-CD-treated cells, randomly chosen from the three independent experiments. *** *p* < 0.05, Student’s *t* test.

**Figure 2 biomedicines-13-00481-f002:**
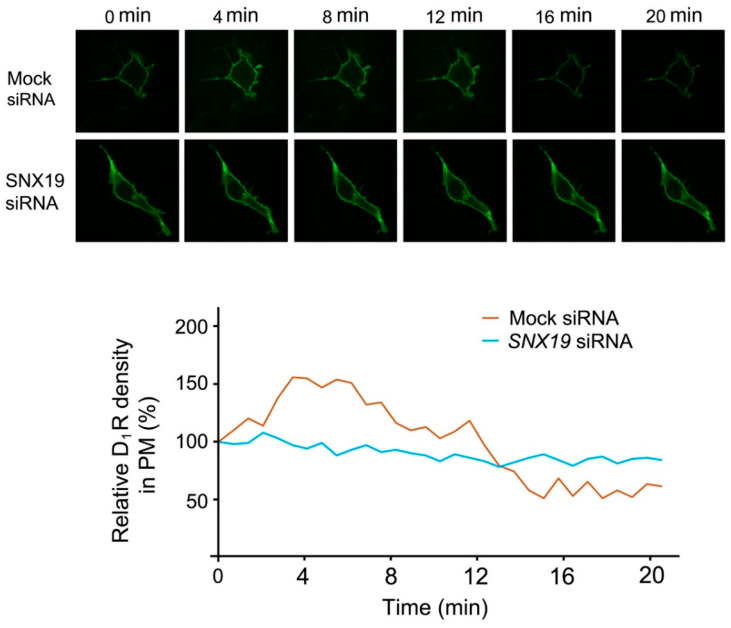
SNX19-mediated D_1_R endocytosis in human RPTCs. Human RPTCs, cultured on glass-bottom microwell dishes, were co-transfected with D_1_R-GFP and scrambled or specific *SNX19* siRNA for 48 hr. The images were captured at the indicated time of FEN treatment, using a spinning disk confocal microscope (Carl Zeiss) and processed using Velocity 6.3 software (PerkinElmer, Waltham, MA, USA). The bottom panel shows the plot of the fluorescence (arbitrary units) of the plasma membrane against time (min). PM, plasma membrane.

**Figure 3 biomedicines-13-00481-f003:**
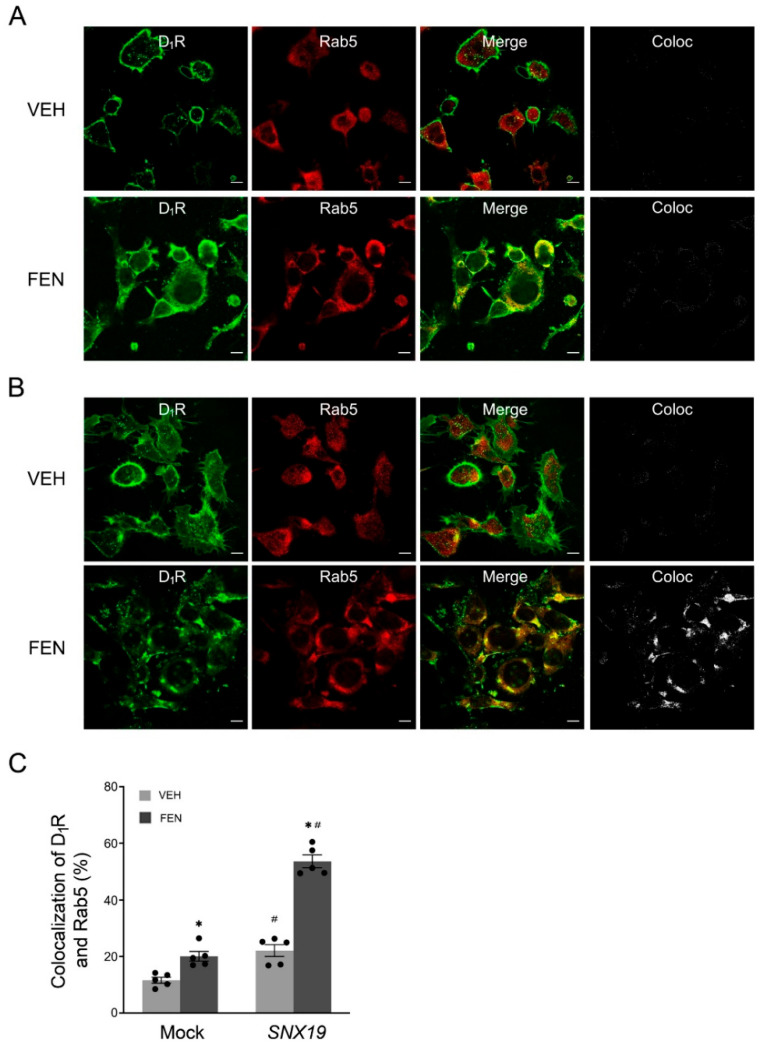
SNX19-mediated D_1_R endocytosis in mouse RPTCs. Mouse RPTCs were transfected with empty-vector (Mock) (**A**) or full-length wild-type SNX19 (**B**) and treated with vehicle (VEH) or fenoldopam (FEN, 25 nM) for 30 min. The RPTCs were stained with anti-D_1_R (green) and Rab5 (red) antibodies. Bar, 10 µm. A panel of separate colocalization images in “Coloc” were generated as described in Methods section. Coloc = colocalization. (**C**) Quantification of the colocalization of D_1_R with Rab5. N = 5, * *p* < 0.05 vs. VEH, ^#^
*p* < 0.05 vs. Mock, two-way ANOVA, Newman–Keuls test.

**Figure 4 biomedicines-13-00481-f004:**
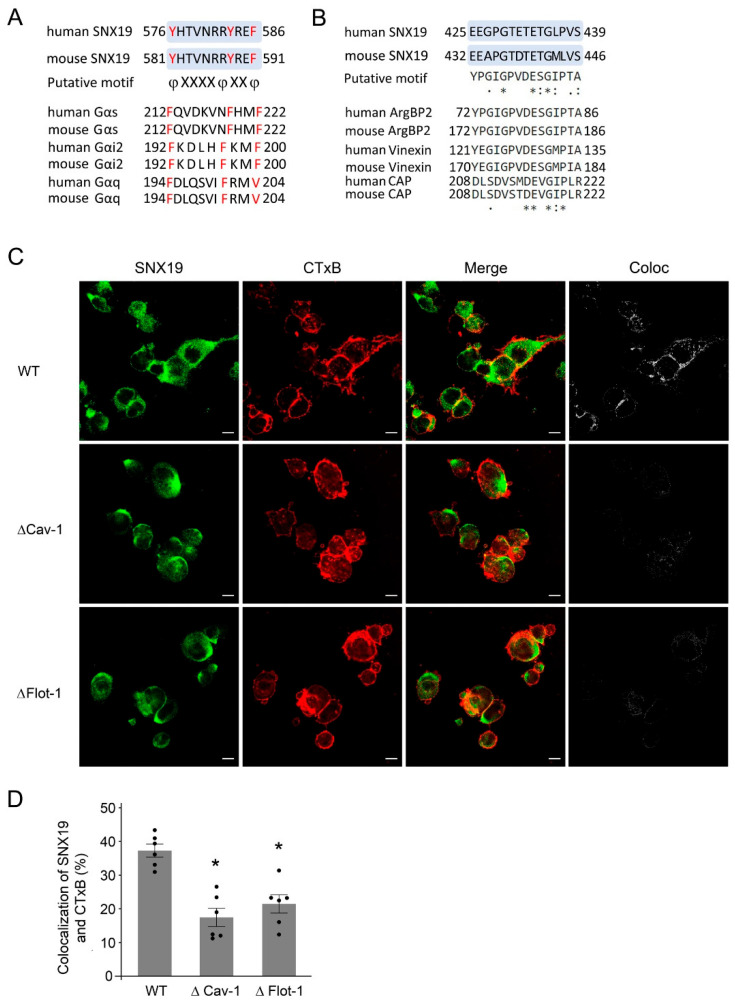
Caveolin-1 and flotillin-1 binding motifs within SNX19. (**A**) Sequence of the caveolin-1 binding motif in SNX19 with homologous regions of human and mouse G proteins with known caveolin-1 binding sites; the red amino acids are the key residues for caveolin-1 binding. The potential canonical and non-canonical caveolin-1 binding motifs in human and mouse SNX19 and G protein sequences are shown. (**B**) Sequence of the flotillin-1 binding motif in SNX19 and alignment with putative SoHo domain, a flotillin-1 binding motif. Three known SoHo domain proteins, human and mouse ArgBP2, vinexin, and CAP are shown as well. Identical amino acids are marked with asterisks (*), conserved amino acids are marked by colon (:), and less-conserved amino acids are marked by dots (.). (**C**) Mouse RPTCs were transfected wild-type (WT-), caveolin-1 (ΔCav1-), or flotillin-1 (ΔFlot1-) binding motif-truncated SNX19 constructs. The RPTCs were fixed with 4% paraformaldehyde and stained with anti-DYK (for SNX19, green) and cholera toxin B-subunit (CTxB, red); yellow in the merged images shows the colocalization of SNX19 and CTxB. Bar, 10 µm. A panel of separate colocalization images in “Coloc” were generated as described in Methods section. Coloc = colocalization. (**D**) Quantification of the colocalization of SNX19 with CTxB in the images from confocal microscopy (**C**), N = 6, * *p* < 0.05 vs. VEH, one-way ANOVA, Newman–Keuls test.

**Figure 5 biomedicines-13-00481-f005:**
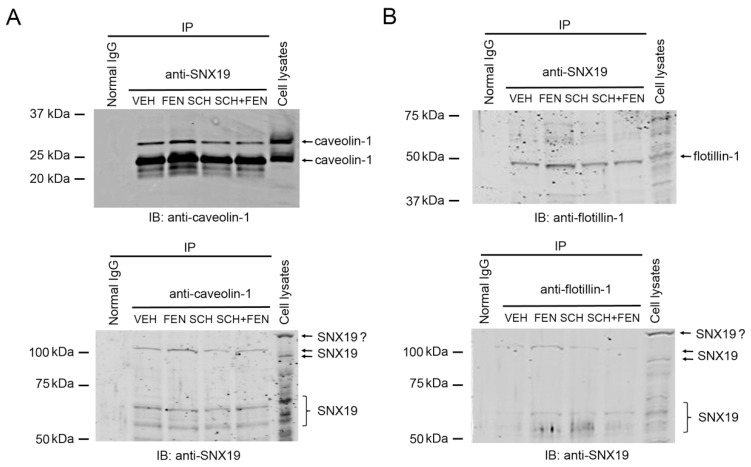
Co-immunoprecipitation of SNX19 with caveolin-1 and flotillin-1. (**A**) Human RPTCs were treated as indicated and the cell lysates were incubated with normal rabbit IgG, anti-SNX19, or anti-caveolin-1 antibodies, as indicated. The precipitated proteins were separated by SDS-PAGE before immunoblotting (IB) with anti-caveolin-1 or anti-SNX19 antibodies, as indicated. (**B**) Cell lysates from the same batch as (**A**) were incubated with normal rabbit IgG, anti-SNX19, or anti-flotillin-1 antibodies, as indicated. The precipitated proteins were separated by SDS-PAGE before immunoblotting (IB) with anti-flotillin-1 or anti-SNX19 antibodies, as indicated. Of note, the displayed molecular sizes are slightly different between the direct immunoblotting and the co-immunoprecipitation of caveolin-1 and SNX19, and the reason for different migration of co-immunoprecipitated proteins are unknown.

**Figure 6 biomedicines-13-00481-f006:**
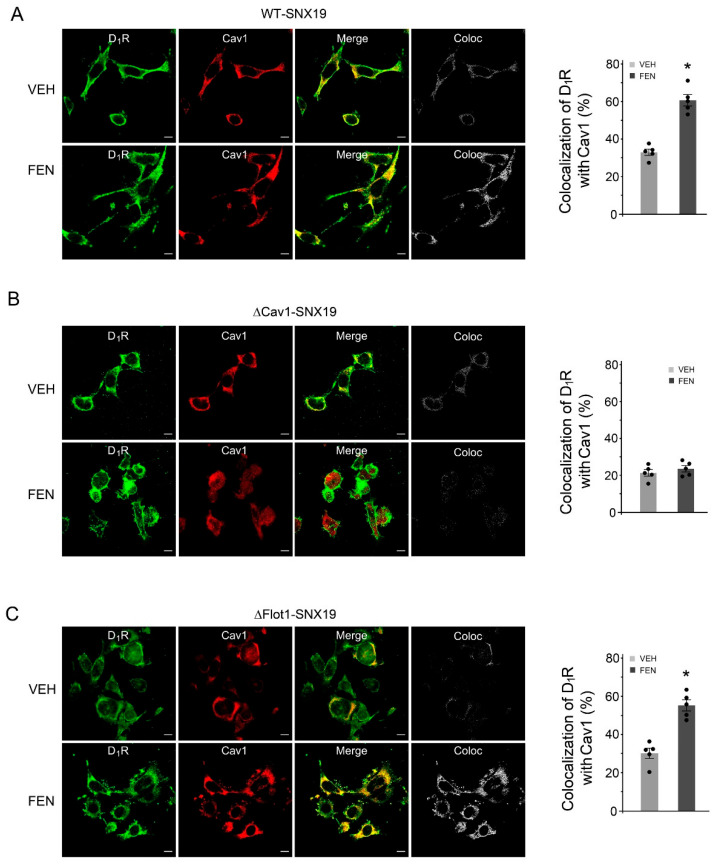
Caveolin1-mediated D_1_R endocytosis in WT-, ΔCav1-, and ΔFlot1-SNX19-transfected mouse RPTCs. (**A**) Mouse RPTCs transfected with wild-type (WT) SNX19 plasmid were treated with vehicle (VEH) or FEN (25 nM) for 30 min. The RPTCs were stained with anti-D_1_R (green) and anti-Cav-1 (red) antibodies. Bar, 10 µm. (**B**) Mouse RPTCs transfected with SNX19 plasmid-deleted caveolin-1 binding motif (ΔCav1-SNX19) were treated and stained as in (**A**). Bar, 10 µm. (**C**) Mouse RPTCs transfected with SNX19 plasmid-deleted flotillin-1 binding motif (ΔFlot1-SNX19) were treated and stained as in (**A**,**B**). Bar, 10 µm. A panel of separate colocalization images in “Coloc” in (**A**–**C**) were generated as described in Methods section. Coloc = colocalization. Quantification of the colocalization of D_1_R with caveolin-1 (Cav1) in images from confocal microscopy (**A**–**C**), N = 5, * *p* < 0.05 vs. VEH, Student’s *t* test.

**Figure 7 biomedicines-13-00481-f007:**
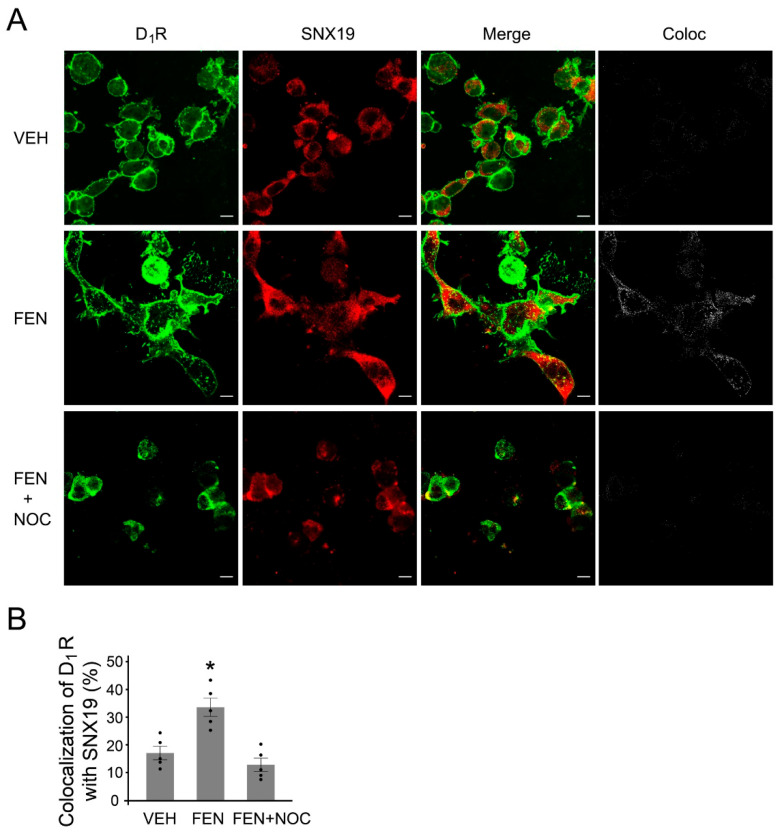
Effect of the inhibition of microtubule polymerization on D_1_R and SNX19 interaction. (**A**) Mouse RPTCs transfected with wild-type SNX19 plasmid were treated with vehicle (VEH) or FEN (25 nM, 30 min) in the absence or presence of nocodazole (NOC, 10 µM, 1 h), a microtubule depolymerization inhibitor. The RPTCs were stained with anti-D_1_R (green) and anti-DYK (for SNX19, red) antibodies. A panel of separate colocalization images in “Coloc” were generated as described in Methods section. Coloc = colocalization. (**B**) Quantification of the colocalization of D_1_R with SNX19 as shown on the right panel. N = 5, * *p* < 0.05 vs. VEH, one-way ANOVA, Newman-Keuls test. Bar, 10 µm.

**Figure 8 biomedicines-13-00481-f008:**
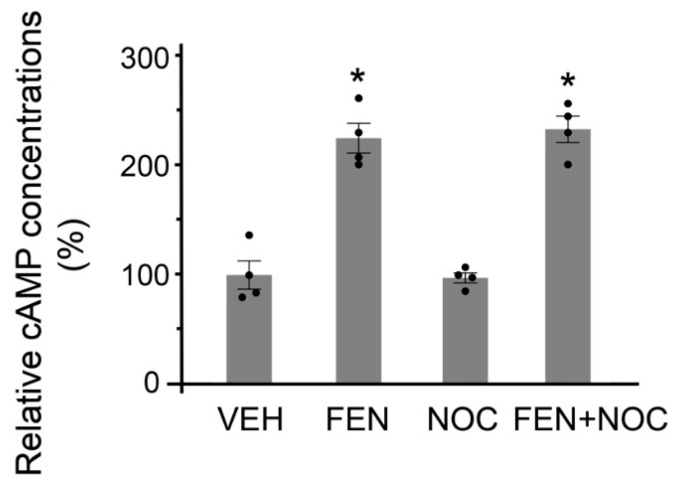
Inhibition of microtubule polymerization does not affect cAMP production. Mouse RPTCs transfected with wild-type SNX19 plasmid were treated with vehicle (VEH) or FEN (25 nM, 30 min) in the absence or presence of nocodazole (NOC, 10 µM, 1 h), a microtubule depolymerization inhibitor. The cAMP concentrations in the supernatant were quantified by ELISA. N = 4, * *p* < 0.05 vs. VEH or NOC, one-way ANOVA, Newman–Keuls test.

## Data Availability

The original contributions presented in this study are included in the article/Appendix A. Further inquiries can be directed to the corresponding author.

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
