# Peer review of "SNX19 Interacts with Caveolin-1 and Flotillin-1 to Regulate D1R Endocytosis and Signaling"

_biomedicines, 2025, doi:10.3390/biomedicines13020481_

Round 1
Reviewer 1 Report
Comments and Suggestions for Authors
The article “SNX19 interacts with caveolin-1 and flotillin-1 to regulate D1R endocytosis and signaling” expands successfully on its subject matter. However, some major concerns need to addressed before the article is ready for publication –
11. The consequences on the downstream gene expression of the SNX19-mediated D1R signalling pathway due to stimulation by D1-like receptor agonist fenoldopam and D1-like receptor antagonist SCH39166 should be validated by western blotting or qPCR with appropriate housekeeping gene expression.
22. The impact of D1R endocytosis and dysregulation on cellular biochemistry should be clearly stated.
33. The relevance and future prospective of this study should be thoroughly discussed.
Author Response
- The consequences on the downstream gene expression of the SNX19-mediated D1R signaling pathway due to stimulation by D1-like receptor agonist fenoldopam and D1-like receptor antagonist SCH39166 should be validated by western blotting or qPCR with appropriate housekeeping gene expression.
Response: It is expected that there are no changes of total caveolin-1 and flotillin-1 expression on the immediate downstream SNX19-mediated D1R endocytosis and signaling with the short-term treatment of fenoldopam (within 30 min) of this study. However, the translocation and redistribution of caveolin-1 and flotillin-1 has been reported. For example, we have reported that D1-like receptor stimulation with fenoldopam (1 µmol/L for 30 min) increases D1R and caveolin-1 translocation in plasm membranes of normal human renal proximal tubule cells (Hypertension 2009, 54:1070). In HEK-293 cells expressing human D1R (D1R-HEK293), fenoldopam increases the expression of caveolin-2 and flotillin-1 in lipid rafts (low density membrane fraction) and association of caveolin-2 (flotillin-1 was not tested) with D1R, effects that were blocked by the D1-like receptor antagonist, SCH23390, which by itself had no effect. Reduction of the expression of caveolin-2 decreases the stimulatory effect on cAMP accumulation (Kidney Int 2004, 66: 2167) in D1R-HEK293 cells. Renal silencing Snx19 in mice decreases the expression of D1R when SNX19 expression is decreased (FASEB J 2020, 34: 6999). This paragraph is now included in the revised version of this manuscript (highlighted in yellow, Line 448-453, Page 15). - The impact of D1R endocytosis and dysregulation on cellular biochemistry should be clearly stated.
Response: The major impact of D1R endocytosis upon activation in the kidney is its regulation of sodium transport and therefore blood pressure. Activation of D1R decreases membrane expression and activities of Na+/K+-ATPase, NHE3, NaPi IIa, sodium bicarbonate cotransporters, and chloride bicarbonate cotransporter in the renal proximal tubule (Am J Physiol 1990, 259: F297; Kidney Int 2000, 57: 534; J Biol Chem 2001; 276: 26906; Am J Physiol Renal Physiol 2007, 292: F230; Hypertension 2009, 54:1070; Kidney Int 2014, 86: 118; PLoS ONE 2018, 13: e0189464), which results in the increase in renal sodium excretion. Therefore, dysregulation of D1R endocytosis leads to aberrant renal sodium excretion and increased blood pressure. The cause of the dysregulation of D1R endocytosis could be the uncoupling of D1R and Gαs (Am J Hypertens 1996, 9: 400; Hypertension 1999, 33: 1036; Proc Natl Acad Sci USA 2002, 99: 3872; Hypertension 2009, 54:1070). The uncoupling of D1R and Gαs has been observed in spontaneous hypertensive rats (Hypertension 2000, 36: 395) and human renal proximal tubule cells from hypertensive subjects (Hypertension 2009, 54: 1070). We have also reported that D1R endocytosis could reduce caveolin-1 deficiency in human renal proximal tubule cells by CAV1 siRNA, and the total cellular expression of caveolin-1 is lower in uncoupling human renal proximal tubule cells than normal human renal proximal tubule cells (Hypertension 2009, 54: 1070). Part of this discussion has now been included in this revised version of this manuscript (highlighted in yellow, Line 468-477, Page 16). - The relevance and future prospective of this study should be thoroughly discussed.
Response: The relevance and future prospective of this study are discussed (highlighted in yellow, Lines 468-477, and 486-496, Page 16).
Reviewer 2 Report
Comments and Suggestions for Authors
SNX19 plays a pivotal role in the localization and trafficking of D1R to lipid raft microdomains. Through an in silico analysis, the authors identified potential binding motifs for caveolin-1 and flotillin-1 within SNX19. The results show that fenoldopam increases the co-localization and co-IP of SNX19 with caveolin-1 and flotillin-1, and are inhibited by a antagonist SCH39166. Moreover, the binding motifs of caveolin-1 and flotillin-1 within SNX19 are critical for D1R endocytosis and signaling. The deletion of these motifs impairs intracellular cAMP production and disrupts the co-localization of SNX19 and D1R. This work highlights the importance of SNX19 in regulating D1R endocytosis through its interaction with caveolin-1 and flotillin-1. While this study has significant implications, I recommend acceptance pending necessary revisions.
Major Comments:
1. Figure 2A: The authors should demonstrate that the SNX19 siRNA effectively reduces the expression of SNX19 with a WB or qPCR analysis to confirm the knock-down efficiency.
2. Section 3.3: The authors show that SNX19 has six residues between the first two aromatic residues, Tyr 576 and Tyr 583, which differs slightly from potassium voltage-gated channel Kv1.3, where there are four residues between the aromatic amino acids Phe (at 216) and Trp (at 221), and is similar to human Gαs, which also has six residues between two aromatic amino acids (Phe at 212 and 219). However, is there any evidence indicating that the non-canonical Gαs motif can bind to caveolin?
3. Figure 5A: In the upper panel, there seems to be a missing stripe, which likely corresponds to the normal IgG control. Additionally, the bands for the indicated proteins in Figure 5 should be clearly marked to improve clarity.
4. Figure 6: The authors demonstrate through fluorescence co-localization that SNX19-mediated D1R endocytosis is dependent on both caveolin-1 and flotillin-1 binding motifs. However, it would be beneficial to complement this data with Co-IP experiments to confirm that the deletion of these motifs disrupts the interaction between SNX19 and caveolin-1/flotillin-1.
Minor Comments:
1. In the abstract, it is not necessary to specify the exact concentrations and durations of Fenoldopam and Nocodazole treatments.
2. In line 228, the citation of reference 14 appears to be inappropriate or misplaced.
Author Response
SNX19 plays a pivotal role in the localization and trafficking of D1R to lipid raft microdomains. Through an in silico analysis, the authors identified potential binding motifs for caveolin-1 and flotillin-1 within SNX19. The results show that fenoldopam increases the co-localization and co-IP of SNX19 with caveolin-1 and flotillin-1, and are inhibited by a antagonist SCH39166. Moreover, the binding motifs of caveolin-1 and flotillin-1 within SNX19 are critical for D1R endocytosis and signaling. The deletion of these motifs impairs intracellular cAMP production and disrupts the co-localization of SNX19 and D1R. This work highlights the importance of SNX19 in regulating D1R endocytosis through its interaction with caveolin-1 and flotillin-1. While this study has significant implications, I recommend acceptance pending necessary revisions.
Major Comments:
- Figure 2A: The authors should demonstrate that the SNX19 siRNA effectively reduces the expression of SNX19 with a WB or qPCR analysis to confirm the knock-down efficiency.
Response: The expression of SNX19 is decreased by specific SNX19 siRNA in human renal proximal tubule cells (Supplementary Figure S2). - Section 3.3: The authors show that SNX19 has six residues between the first two aromatic residues, Tyr 576 and Tyr 583, which differs slightly from potassium voltage-gated channel Kv1.3, where there are four residues between the aromatic amino acids Phe (at 216) and Trp (at 221), and is similar to human Gαs, which also has six residues between two aromatic amino acids (Phe at 212 and 219). However, is there any evidence indicating that the non-canonical Gαs motif can bind to caveolin?
Response: Both 4 and 6 amino acids between two aromatic residues in canonical and non-canonical binding motifs can bind to caveolin-1. Gas with 6 amino acids between two aromatic amino acids (Phe) has been shown to bind to caveolin-1 by GST pull-down experiments (J Biol Chem 1995, 270: 15693). Gaq also with 6 amino acids between two aromatic amino acids (Phe) has been shown to bind to caveolin-1 by fluorescence resonance energy transfer and co-immunoprecipitation (J Cell Sci 2008, 121: 1363). Gai2 has 4 amino acids between two aromatic amino acids (Phe), which has been shown to bind to caveolin-1 by GST pull-down experiments (J Biol Chem 1995, 270: 15693). SNX19, like Gαq, has a non-canonical caveolin-1 binding motif that interacts with caveolin-1. Whether or not the putative caveolin-1 binding motif within SNX19 is strictly required for its binding to caveolin-1 or caveolin-1 binds to another region of SNX19 warrants further investigation. This is discussed in the revised manuscript (highlighted in yellow, Lines 394-402, Pages 14). - Figure 5A: In the upper panel, there seems to be a missing stripe, which likely corresponds to the normal IgG control. Additionally, the bands for the indicated proteins in Figure 5 should be clearly marked to improve clarity.
Response: Thank you very much for pointing out this matter. The negative IgG control is shown in Figure 5 in this revised manuscript. Per this reviewer’s suggestion, the bands are indicated with arrows in the revised Figure 5. Of note, the displayed molecular sizes are slightly different between the direct immunoblotting and the co-immunoprecipitation of caveolin-1 and SNX19. However, the reasons for the different migration of co-immunoprecipitated proteins are unknown. - Figure 6: The authors demonstrate through fluorescence co-localization that SNX19-mediated D1R endocytosis is dependent on both caveolin-1 and flotillin-1 binding motifs. However, it would be beneficial to complement this data with Co-IP experiments to confirm that the deletion of these motifs disrupts the interaction between SNX19 and caveolin-1/flotillin-1.
Response: We agree with the reviewer that it will make our observation more solid and will help to study potential differences (or similarity) of their association between humans and mice in the future. We have previously performed the reviewer suggested co-IP experiments, unfortunately, all our co-IP experiments were not successful in mouse renal proximal tubule cells transfected with wild-type or mutant SNX19 plasmids. The exact reasons are not known. It is possible that the affinity of anti-SNX19 is not high enough to bind caveolin-1 or flotillin-1 or some other factors that prevent their binding to each other (SNX19 and caveolin-1 or flotillin-1) in mouse renal proximal tubule cells.
Minor Comments:
1. In the abstract, it is not necessary to specify the exact concentrations and durations of Fenoldopam and Nocodazole treatments.
Response: We have removed the concentrations of fenoldopam and nocodazole in the abstract in this revised manuscript.
2. In line 228, the citation of reference 14 appears to be inappropriate or misplaced.
Response: We have removed Ref. 14 in Line 234, Page 7 (Line 228, Page 7) in this revised manuscript. It is not misplaced, because Ref. 14 mainly investigates D1R palmitoylation at C347 and C351 by SNX19 and in vivo blood pressure regulation by SNX19 lipid raft distribution, which does not directly study D1R trafficking regulated by SNX19.
Round 2
Reviewer 1 Report
Comments and Suggestions for Authors
The authors have successfully addressed the queries. The article is suitable for publication.
Author Response
We appreciate the reviewer’s positive comments.
Reviewer 2 Report
Comments and Suggestions for Authors
The authors have satisfactorily addressed all my questions, and I have no more comments.
Author Response

(The authors gave the same response as above.)
